# Nitrate Signaling and Its Role in Regulating Flowering Time in *Arabidopsis thaliana*

**DOI:** 10.3390/ijms25105310

**Published:** 2024-05-13

**Authors:** Mengyun Wang, Jia Wang, Zeneng Wang, Yibo Teng

**Affiliations:** 1College of Life and Environmental Sciences, Hangzhou Normal University, Hangzhou 311121, China; mengyunwang@zju.edu.cn (M.W.);; 2College of Life Sciences, Zhejiang University, Hangzhou 310058, China; 3Kharkiv Institute, Hangzhou Normal University, Hangzhou 311121, China

**Keywords:** Arabidopsis, nitrate signaling, flowering time, calcium signaling, NLP7

## Abstract

Plant growth is coordinated with the availability of nutrients that ensure its development. Nitrate is a major source of nitrogen (N), an essential macronutrient for plant growth. It also acts as a signaling molecule to modulate gene expression, metabolism, and a variety of physiological processes. Recently, it has become evident that the calcium signal appears to be part of the nitrate signaling pathway. New key players have been discovered and described in *Arabidopsis thaliana* (Arabidopsis). In addition, knowledge of the molecular mechanisms of how N signaling affects growth and development, such as the nitrate control of the flowering process, is increasing rapidly. Here, we review recent advances in the identification of new components involved in nitrate signal transduction, summarize newly identified mechanisms of nitrate signaling-modulated flowering time in Arabidopsis, and suggest emerging concepts and existing open questions that will hopefully be informative for further discoveries.

## 1. Introduction

Nitrogen (N) is one of the essential macronutrients in plants. In the soil ecosystem, N exists in many different forms, including both inorganic (e.g., ammonium, nitrate) and organic forms derived by fertilizer or N fixation by legume plants. Nitrate is the major source of N in many soil systems. After being taken up by plant roots, nitrate is reduced to ammonium for further synthesis into cellular components such as nucleic acids, amino acids, proteins, and other cellular components necessary for their growth and development. In addition to its role as a nutrient, nitrate has been shown to act as a signaling molecule in plants [1,2,3]. It can trigger signaling pathways that regulate gene expression, metabolism, and many physiological processes [4,5,6,7]. Thus, the role of nitrate serving as a signaling molecule contributes greatly to morphological changes in plants [4].

In the currently established nitrate signaling model, NITRATE TRANSPORTER 1.1 (NRT1.1), located in the plasma membrane, is able to sense the fluctuation in nitrate levels. Upon nitrate levels being elevated, it activates the calcium-channel, resulting in a rapid influx of calcium (Ca^2+^) into the cytoplasm. This nitrate-coupled Ca^2+^ signal triggers downstream responses to alterations in N nutrient availability. In these downstream responses, the typical transcriptional regulation pathway is dependent on the NIN (nodule inception)-like proteins (NLPs), particularly NLP6 and NLP7, which can bind directly to many nitrate-responsive genes and stimulate their transcription (Figure 1). A more detailed description of the components in the nitrate signaling cascade and their roles are listed in Table 1.

During a fluctuation in nitrate levels in the soil environment, plants are able to integrate transcriptome and cellular metabolism with developmental patterning in order to adapt to environmental changes by shaping organ biomass and architecture. Among architectural features, the root system is the most significantly affected by N availability. For example, when *Arabidopsis thaliana* (Arabidopsis) seedlings are grown under severe N deficiency conditions, it restricts root branching and elongation. While mild N deficiency stimulates root elongation and branching [8]. A large amount of knowledge has accumulated regarding the signaling events occurring during the root adaptation to altered N supply, which have been extensively covered by excellent reviews [8,9,10,11,12]. Here, we specifically focus on the systematic regulation of flowering time by N signaling.

**Table 1 ijms-25-05310-t001:** Components involved in nitrate signaling.

Gene Name	Locus	Protein Identity	Roles in Nitrate Signaling	Functions	Reference
*NRT1.1*	AT1G12110	NITRATE TRANSPORTER 1.1 (NRT1.1)	Plasma membrane nitrate transceptor	Sensing a wide range of external nitrate concentrations and generating different levels of the primary nitrate response (PNR)	[1]
*CNGC15*	AT2G28260	CYCLIC NUCLEOTIDE-GATED CHANNEL 15 (CNGC15)	Nitrate-specific calcium channel	Dynamically interacting with NRT1.1 to control the calcium influx in a nitrate-dependent manner	[13]
*CPK10* (*CDPK1*)	AT1G18890	CALCIUM-DEPENDENT PROTEIN KINASE 1 (CDPK1)	Nitrate-coupled calcium sensor	Phosphorylating NLP7/NLP6 to facilitate the cytoplasm-nuclear shuttling	[3]
*CPK30*	AT1G74740	CALCIUM-DEPENDENT PROTEIN KINASE 30 (CPK30)	Nitrate-coupled calcium sensor	Phosphorylating NLP7/NLP6 to facilitate the cytoplasm-nuclear shuttling	[3]
*CPK32*	AT3G57530	CALCIUM-DEPENDENT PROTEIN KINASE 32 (CPK32)	Nitrate-coupled calcium sensor	Phosphorylating NLP7/NLP6 to facilitate the cytoplasm-nuclear shuttling	[3]
*NLP7*	AT4G24020	NIN LIKE PROTEIN 7 (NLP7)	Intracellular nitrate sensor; Master transcription factor of PNR	Binding nitrate via its amino terminus, which results in the derepression of NLP7 as a transcription activator; Binding the nitrate-responsive cis-element (NRE) to activate nitrate-responsive gene expression	[2,14,15,16,17,18,19]
*NLP6*	AT1G64530	NIN-LIKE PROTEIN 6 (NLP6)	Transcription factor	Acting as a PNR regulator partially redundant with NLP7	[2,20,21]
*NLP2*	AT4G35270	NIN-LIKE PROTEIN 2 (NLP2)	Transcription factor	Acting as a major PNR regulator	[2,21,22]
*NLP4*	AT1G20640	NIN-LIKE PROTEIN 4 (NLP4)	Transcription factor	Participating in the regulation of PNR	[2,21]
*NLP5*	AT1G76350	NIN-LIKE PROTEIN 5 (NLP5)	Transcription factor	Participating in the regulation of PNR	[2,21]
*NLP8*	AT2G43500	NIN-LIKE PROTEIN 8 (NLP8)	Transcription factor	Participating in the regulation of PNR	[2,21]
*NLP9*	AT3G59580	NIN-LIKE PROTEIN 9 (NLP9)	Transcription factor	Participating in the regulation of PNR	[2]

In the currently accepted flowering model of Arabidopsis, several genetic pathways have been considered: the photoperiod pathway, the gibberellin (GA) pathway, the vernalization pathway, the autonomous pathway, the endogenous/age pathway, and the thermosensory pathway [23,24]. These pathways integrate endogenous and external cues into downstream effectors known as flowering pathway integrators, such as FLOWERING LOCUS T (FT), SUPPRESSOR OF OVER EXPRESSION OF CONSTANS (SOC1), and LEAFY(LFY), to ensure flowering [25,26,27,28,29,30]. Flowering signals are translated into floral meristems at the flanks of the shoot apical meristem (SAM) through the mobile protein FT. Once in the SAM, FT forms a complex with several partners, causing changes in gene expression of downstream flowering components that reprogram the SAM to form flowers [31,32].

As an essential nutrient to support plant growth, N availability strongly influences flowering time [33,34]. A U-shaped flowering curve resulting from different N levels has been proposed [35]. There is an optimal concentration of nitrate to stimulate flowering. When the concentration is above or below this, flowering is retarded. Studies have suggested that nitrate serves as both a nutrient source and as a signaling molecule for floral induction [36,37,38]. Here we summarize the role of nitrate on flowering time from the perspective of signaling function. Nitrate signaling has been suggested to regulate flowering time through different flowering pathways, depending on nitrate treatment and concentration (Figure 2).

In this review, we summarize recent research achievements regarding the perception and transduction of nitrate signals based on studies performed using the model plant Arabidopsis. We analyze the nitrate signaling cascade from transceptor to key transcription factors and discuss how these critical components play their roles. In addition, we examine the relationship between nitrate availability and flowering time and highlight the role of nitrate signaling in modulating flowering time.

## 2. Nitrate Signal Perception

### 2.1. Nitrate Sensing in the Plasma Membrane

NITRATE TRANSPORTER 1.1 (NRT1.1) is a receptor on the plasma membrane that detects changes in soil nitrate concentrations in Arabidopsis [1]. Initially characterized as a dual-affinity transporter involved in root uptake of nitrate [39], the *nrt1.1* mutant exhibited reduced growth of nascent roots, stems, leaves, and flower buds, as well as a late-flowering phenotype [40]. Additionally, nitrate-dependent gene transcription, lateral root growth, and seed germination were changed in a *nrt1.1* mutant [41,42,43,44]. The phenotype provides useful cues for the crucial role of NRT1.1 in plant development. Then the question arising at that time was whether these defects in the *nrt1.1* mutant were caused by reduced nitrate uptake or the involvement of NRT1.1 in nitrate signaling. To address this question, Ho et al. used an uptake- and sensing-decoupled mutant, *chl1-9*, and discovered that the sensor function of NRT1.1 is independent of its uptake activity. Plants are able to sense changes in nitrate concentration through the dual affinity binding of NRT1.1 and a phosphorylation switch at residue threonine 101 [1]. Since the discovery of NRT1.1 as a nitrate receptor, numerous studies have focused on NRT1.1-mediated nitrate sensing, providing evidence for the role of NRT1.1 in N signaling. This has been described extensively in several reviews [45,46].

Prior to the discovery of NRT1.1 as a nitrate receptor, it was suggested as early as 1997 that calcium (Ca^2+^) might be involved in nitrate signaling as a second messenger [47]. Several nitrate-induced genes were altered by pretreatment with the Ca^2+^ chelator ethylene-bis (oxyethylenenitrilo) tetraacetic acid or the Ca^2+^ channel blocker lanthanum chloride in maize, barley, and Arabidopsis [47,48,49]. These results strongly implicate an interplay between the nitrate response and calcium-related signaling pathways. Moreover, it was found that nitrate treatments transiently increased cytosolic Ca^2+^ concentrations, but this increase was abolished in *nrt1.1* mutant seedlings with the facilitation of aequorin reporter plants [48]. Similarly, a unique and dynamic nuclear and cytosolic Ca^2+^ signature was shown to be specifically stimulated by nitrate using the ultrasensitive Ca^2+^ biosensor GCaMP6 expressed in single cells of mesophyll protoplasts [3]. Based on these early findings on the involvement of the Ca^2+^ signature in nitrate signaling, it raised the question of which component modulates the entry of Ca^2+^ ions into the cytoplasm coupled with nitrate signaling. Until 2021, the Ca^2+^ channel specific for the nitrate-induced Ca^2+^ spike in Arabidopsis was not identified. A cyclic nucleotide-gated channel (CNGC) protein, CNGC15, can interact with the nitrate transceptor NRT1.1 to regulate Ca^2+^ influx. When nitrate levels are low, NRT1.1 and CNGC15 form a complex to block the Ca^2+^ channel. Upon nitrate levels elevating, the NRT1.1-CNGC15 complex displays calcium-channel activity, resulting in nitrate-induced Ca^2+^ influx [13]. This important discovery provides direct evidence that Ca^2+^ is a second messenger in the Arabidopsis nitrate signaling pathway.

### 2.2. Nitrate Sensing in the Cytoplasm

Plants can also sense nitrate directly in the cytoplasm to adjust their metabolic and growth responses. The NIN (nodule inception)-like proteins (NLPs) are crucial transcription factors of the nitrate signaling pathway, initiating nitrate-induced transcriptional changes [50,51,52]. In addition to being a transcription factor, the NLP7 protein also acts as a nitrate sensor. It functions within the cytoplasm, which is distinct from NRT1.1 functioning on the plasma membrane [2]. Thus, plants possess two spatially different mechanisms for sensing nitrate, both intracellular and extracellular.

Using two Biomolecule Interaction Analyzers (microscale thermophoresis and surface plasmon resonance), it was shown that nitrate can bind directly to NLP7 [2]. In addition to the in vitro assays, a genetically encoded fluorescent split biosensor, mCitrine-NLP7, was developed, which acts as a fluorescent nitrate receptor in vivo [2]. This facilitates the visualization of nitrate dynamics in single cells in plants. 10 mM nitrate, but not potassium chloride, could induce a Citrine fluorescence signal in both mesophyll cells and primary root tip cells. Furthermore, the fluorescence signal was also detected in plants under a wide range of nitrate concentrations, from 100 µM to 10 mM [2]. As the nitrate biosensor is very sensitive and specific for nitrate, the real-time live imaging observation strongly supports that NLP7 binds to nitrate. The sensing mechanism is that NLP7 can perceive nitrate via its amino terminus, causing it to undergo a conformational change and to be transcriptionally derepressed [2].

## 3. Nitrate Signal Transduction, from the Extranuclear to the Nucleus

### 3.1. Calcium-Binding Proteins

Calcium (Ca^2+^) has been well established as a second messenger in plants, participating in many signal transduction processes, especially in response to environmental alterations [53,54,55]. A spatio-temporal pattern of Ca^2+^ could rise through channels, exporters, or pumps triggered by different environmental cues. This specific alteration of Ca^2+^ levels can be detected and decoded by Ca^2+^-binding proteins. These include calmodulin (CaM), calmodulin-like proteins (CMLs), Ca^2+^-dependent protein kinases (CPKs), and calcineurin B-like proteins (CBLs) and their interacting kinases (CIPKs) [56]. These Ca^2+^-binding proteins bridge cytosolic Ca^2+^ elevations to downstream responses.

While NITRATE TRANSPORTER 1.1 (NRT1.1) perceives the nitrate change signal, an influx of Ca^2+^ into the cytoplasm is produced synergistically through the transceptor-channel complex. To translate nitrate-coupled cytosolic Ca^2+^ elevations into downstream actions, CPK10, CPK30, and CPK32 protein kinases have been identified as the main decoders that regulate universal nitrate-inducible responses in Arabidopsis [3]. The three kinases effectively activate the luciferase reporter gene driven by the nitrate marker gene promoter in response to nitrate in the protoplast system and control the primary nitrate-responsive transcriptome. The *cpk10 cpk30 cpk32* triple mutant displayed defects in the primary nitrate-mediated shoot developmental program and root system architecture. Additionally, CIPK8 and CBL7 were found to be involved in a kinase cascade in nitrate signaling [57,58]. Nitrate rapidly induced *CIPK8* expression. The *cipk8* mutant was defective in the low-affinity phase of the primary nitrate response and nitrate-modulated primary root growth. CBL7 regulates the low-nitrate response in Arabidopsis.

### 3.2. The Nin like Proteins—Primary Nitrate Response Pathway

NIN LIKE PROTEIN 7 (NLP7) from Arabidopsis is one of the most extensively studied NLPs in plants. Arabidopsis *nlp7* mutant plants displayed an N-starved phenotype, and the expression profiling of nitrate uptake, assimilation, and response genes in the *nlp7* mutant was impaired [14]. Additionally, nitrate can induce the nuclear accumulation of the NLP7 protein [15]. Therefore, it was proposed that NLP7 is a key transcription factor involved in the regulation of nitrate assimilation in higher plants. As a key transcription factor, the process of nuclear translocation is a crucial step. A phosphorylation-mediated nuclear cytoplasmic shuttling mechanism for NLP7 translocation into the nucleus has been discovered, with CPKs playing a key role in phosphorylating NLP7 [3]. When the nitrate-induced increase in Ca^2+^ concentration was blocked by Ca^2+^ chelators or channel blockers, the nitrate-stimulated NLP7 phosphorylation by CPK10, CPK30, and CPK32 was greatly diminished. Furthermore, when the CPK phosphorylation site in NLP7, a uniquely conserved serine (Ser205), was mutated to alanine (S205A), the phosphorylation also failed. In addition, the *cpk10 cpk30 cpk32* triple mutant plants also lost nitrate-stimulated phosphorylation. As a result, the expression of nitrate-responsive genes was markedly diminished, and nitrate-specific stimulation of the lateral root was severely retarded. This is due to the failure of the phosphorylation resulting from the absence of these Ca^2+^-binding protein kinases, which limits the nuclear translocation of NLP7. Consequently, it can’t regulate the gene expression response to nitrate. In addition, the nitrate-stimulated nuclear translocation of NLP7 was strongly blocked in the *cyclic nucleotide-gated channel 15* (*cngc15*) mutant plants due to impaired Ca^2+^ signals [13]. This demonstrates that nitrate-dependent translocation of NLP7 is calcium signaling-dependent.

Once in the nucleus, NLPs can bind directly to nitrate-responsive cis-elements (NRE) in the promoter regions of nitrate-responsive genes and activate transcription [16]. The NRE is a conserved 43 bp sequence first identified in the promoter of the Arabidopsis nitrite reductase gene (*NIR1*), and the motif exists in the promoters of *NIR1* from various higher plants [17,18,19]. Nitrate can induce numerous genes expression within minutes, which peak at around 30 min. This rapid transcriptional response is termed the primary nitrate response (PNR) [59]. Typical PNR genes include those involved in nitrate absorption or assimilation. NRE has been identified in many PNR genes. Genome-wide analyses revealed that NLP7 binds to the promoters of many PNR genes involved in N metabolism, such as transporters, enzymes, and transcription factors [15,60]. The expression of PNR genes is impaired in the absence of NLP7. It is believed that NLP7 acts as a master regulator of PNR [14,60]. NLP6, the closest homolog of NLP7, has redundant roles. Knockout mutations of *NLP6* also result in reduced induction of many PNR target genes [20]. The transcriptome analysis of the PNR showed that NLP6 and NLP7 govern approximately 50% of nitrate-induced genes, with both distinct and partially functionally redundant patterns [21]. Other NLP family members, such as NLP2, -4, -5, -6, and -8, also participated in regulating the PNR, especially NLP2, which serves as the major transcription factor as NLP7 to regulate early nitrate responses by binding hundreds of genes related to the transcriptional response to nitrate [21,22]. Thus, NLPs-dependent pathways play a crucial role in mediating nitrate signal transduction from the extranuclear to the nucleus to regulate downstream nitrate-induced responses.

Numerous studies have demonstrated the significant role of NLPs in regulating plant growth dependent on N signaling. For example, NLP7 directly binds to the promoter of *TRYPTOPHAN AMINOTRANSFERASE RELATED 2* (*TAR2*), a tryptophan aminotransferase-related gene, and activates its expression, thereby regulating lateral root development [61]. Additionally, NLP7 modulates auxin pathways to regulate root cap development [62]. Similarly, NLP8 directly activates the expression of *CYTOCHROME P450* (*CYP707A2*), an ABA catabolic enzyme, to regulate nitrate-promoted germination [63]. There are still a large number of genes that are likely to be directly controlled by NLP activity. A comprehensive analysis of NLP target genes would establish the central network of NLP-PNR pathway that bridges PNR genes to physiological processes. This deserves further exploration.

## 4. Flowering Time Response to Nitrate Availability

### 4.1. Signal to the Photoperiod Pathway to Regulate Flowering

As a long-day (LD) plant, Arabidopsis relies on the photoperiod pathway to promote flowering. The central regulator in photoperiodic flowering is CONSTANS (CO), which directly binds to the promoter of *FLOWERING LOCUS T* (*FT*) and activates its expression [64]. It was demonstrated that nitrogen (N) conditions may affect the transcriptional levels of *CO* [33,36,65]. Specifically, they were higher in the low-N conditions than those in the high-N conditions [36,65]. This suggests that CO may be one of the crucial components connecting N signaling with the flowering pathway.

One of the major features of CO is that it exhibits a diurnal expression pattern that is under the control of the circadian clock [66]. The central circadian components, such as CIRCADIAN CLOCK ASSOCIATED 1 (CCA1), LATE ELONGATED HYPOCOTYL 1 (LHY), and TIMING OF CAB EXPRESSION 1 (TOC1), influence the expression of downstream targets to regulate many aspects of plant growth, such as floral transition [67,68,69]. CO is controlled by several circadian clock-associated proteins at the transcriptional or post-translational level [70,71], and a genetic linkage between the central clock circuitry and the downstream CO-FT photoperiodic flowering pathway has been verified [72]. Thus, CO-FT photoperiodic flowering has been considered a circadian output. It has long been proposed that N signals may represent an input mechanism to affect plant circadian clock function [73]. Consistent with this, low-N conditions (1.94 mM nitrate/1 mM ammonia) increase the expression of key components of the central oscillator, *CCA1*, *LHY*, and *TOC1*, while high-N conditions (77.6 mM nitrate/46 mM ammonia) decrease the expression of these genes. As a circadian output, the transcriptional levels of *CO* were higher in the low-N conditions synchronously [65]. Yuan et al. (2016) also identified the components that bridge N signals to the central clock: cryptochrome blue-light receptor 1 (CRY1) and FERREDOXIN-NADP (+) OXIDOREDUCTASE (FNR1). FNR1 acts as an upstream positive regulator of CRY1, influencing the nuclear phosphorylation and degradation of CRY1 [65]. The abundance of cellular CRY1 protein plays a role in circadian clock input [74,75,76,77]. Therefore, it is possible that N functions as an input signal to the central circadian clock via CRY1 to mediate flowering time. In summary, the N signaling pathway regulates the abundance of cellular CRY1 protein, which in turn impacts the expression of crucial components of the central oscillator. As a result, the expression levels of *CO*, which are responsible for photoperiodic flowering activation, are altered, ultimately affecting the timing of flowering.

Additionally, there is a circadian clock-independent way that N levels regulate flowering time through the photoperiod pathway. The FLOWERING BHLH (FBH) transcription factor family plays a crucial role in activating *CO* gene transcription. They can directly bind to the E-box elements in the *CO* promoter and activate its transcription. In addition, the FBH proteins increase *CO* levels throughout the day without affecting their diurnal pattern [78]. Sanagi et al. identified FBH4 as a crucial regulator of N-responsive flowering in Arabidopsis [36]. The phosphorylation state of the FBH4 protein can be modulated by N conditions. Under low N conditions, FBH4 phosphorylation levels decrease, promoting FBH4 nuclear localization and transcriptional activation of the direct target *CO* and downstream florigen *FT*. Thus, FBH4 plays an equally important role in N signaling by interacting with the CO-FT photoperiodic pathway to modulate flowering time.

### 4.2. Crosstalk with Gibberellin Signaling in Flowering Regulation

The plant hormone gibberellin (GA) regulates various aspects of plant growth and development in response to environmental signals [79,80,81]. Bioactive GA promotes growth by counteracting the functions of DELLA growth repressing proteins (DELLAs) [82]. In the absence of GA, DELLA acts as a negative regulator of plant growth by altering the activity of transcription factors or regulators of GA-responsive genes through protein-to-protein interaction. After bioactive GAs bind to their receptor, GIBBERELLIN INSENSITIVE DWARF 1 (GID1), they promote GID1 to interact with DELLA protein, resulting in the destruction of DELLA protein via the ubiquitin/26S proteasome-dependent pathway. This releases the inhibitory interaction of DELLA protein with TFs, thereby promoting plant growth. GA is critical for flowering in Arabidopsis, promoting flowering by activating the floral integrator genes [83,84,85]. Exogenous treatment with GA compounds accelerated flowering time in Arabidopsis; the GA biosynthesis or receptor mutant plants showed a delayed flowering phenotype; the late-flowering mutant plants were impaired in GA biosynthesis or signaling [86,87,88,89,90].

It has been reported that nitrate modulates flowering time through the GA pathway [33,91]. Compared to Arabidopsis seedlings grown under 3 mM nitrate, the expression levels of bioactive GA biosynthesis-related genes, such as *GIBBERELLIN 20-OXIDASE 1* (*GA20OX1*), *GIBBERELLIN 3-OXIDASE 1* (*GA3OX1*), were higher when grown under 1 mM nitrate. Consistent with this, DELLA protein levels were lower under 1 mM nitrate. This means that seedlings grown under 1 mM nitrate have higher levels of bioactive GA compared to 3 mM nitrate, thereby promoting the degradation of DELLAs. The reduced DELLA accumulation, in turn, promotes flowering time. Another strong evidence is that the loss-of-function mutant plants of DELLA protein had the same flowering time when grown under either 1 mM or 3 mM nitrate. This suggests that higher levels of nitrate repress bolting through the accumulation of DELLA proteins [91].

GATA TRANSCRIPTION FACTOR 21 (GNC) and GATA TRANSCRIPTION FACTOR 22 (GNL/CGA1) are GATA-type transcription factors downstream of DELLA proteins that repress GA signaling [92], thereby controlling many aspects of plant development through growth regulatory signals [93]. It was found that nitrate-regulated flowering time is also dependent on the two transcription factors. Compared to Arabidopsis seedlings grown under 1 mM nitrate, the expression levels of *GNC* and *CGA1/GNL* were higher under 3 mM nitrate, consistent with the accumulation of DELLA proteins. Consequently, GNC and CGA1/GNL positively influence the expression of *SCHLAFMUTZE* (*SMZ*) and *SCHNARCHZAPFEN* (*SNZ*) [91]. SMZ and SNZ are two AP2 family transcription factors, acting as floral repressors. SMZ binds directly to the *FT* promoter and inhibits its expression [94,95]. When grown under higher nitrate, the transcriptional levels of the *SMZ* and *SNZ* are relatively higher due to the higher GNC and CGA1/GNL, which in turn delays flowering time [91].

Thus, here is another way for nitrate signaling to regulate flowering time: nitrate availability controls flowering time by modulating *SMZ* and *SNZ* gene expression, which is dependent on the GA pathway. Future research is required to clarify the mechanistic details of how nitrate signaling is involved in regulating the GA pathway.

### 4.3. Nitrate Signaling Directly Integrated into the Age Pathway in the Shoot Apical Meristem

Signaling events occurring in the shoot apical meristem (SAM) are crucial for successful reproduction [23]. Environmental or endogenous signals are integrated into the SAM, allowing fine-tuning for developmental responses. For example, plants can sense the availability of nitrate in the soil to control their organogenesis rate through long-range signaling by cytokinin hormone precursors that travel through the plant [96]. Trehalose-6-phosphate signaling affects flowering time directly in the SAM [97,98]. Genes related to sugar and hormone signaling and response were found to be involved in tissue specific events that control flowering in the SAM [99]. Most of all, signals in flowering pathways are initiated in the leaves and then transported to the SAM by the mobile FT proteins [31,100]. Once in the SAM, FT forms a complex with several partners and directly up-regulates large amounts of genes to induce the formation of flowers [32,101].

It has been suggested that nitrate signaling regulates flowering via the SQUAMOSA PROMOTER BINDING PROTEIN-LIKE (SPL) transcription factors [97]. The SPL family has the ability to promote flowering and functions in the endogenous/age pathway [102,103]. It shows a high degree of functional redundancy among its members. Eight members of the Arabidopsis SPL family, SPL2, SPL3, SPL4, SPL5, SPL9, SPL10, SPL11, and SPL15, have been implicated in the floral transition [24,104]. Among them, SPL3 and SPL5 were found to connect nitrate signaling to the endogenous flowering network in the SAM [97]. Functional nitrate-responsive cis-elements (NRE) motifs were identified in *SPL3* and *SPL5*. Meanwhile, NIN LIKE PROTEIN 6 (NLP6) and NIN LIKE PROTEIN 7 (NLP7) are expressed in the SAM. In vitro assays provide evidence that the NRE motifs present in the upstream intergenic regions of *SPL3* and *SPL5* can activate transcription. SUPPRESSOR OF OVER EXPRESSION OF CONSTANS (SOC1) is one of the flowering integrators that acts as a central integrator for multiple flowering pathways in the SAM [28]. Since *SOC1* expression was greatly reduced in the plants grown under limited nitrogen supply [97], it was postulated that SOC1 may be the main flowering integrator in nitrate-dependent flowering in the SAM. In summary, plants can directly sense the nitrate status and mediate flowering at the SAM through the nitrate-responsive factors NLP7 and NLP6, which directly bind to the promoters of *SPL3* and *SPL5* to regulate at the transcriptional level. Crosstalk between nitrate and phytohormone or sugar signaling pathways has been considered to modulate plant developmental programs [105,106,107]. The signaling overlaps between nitrate and hormones involved in root development have been largely investigated [108], but much less in the reproductive process. Therefore, it remains intriguing to elucidate the mechanistic details of how plants tissue-specifically integrate nitrate signaling with endogenous signals into floral pathway-related genes to mediate bolting time in the SAM.

## 5. Perspective

Significant advances have been made in the field of the nitrate signaling pathway and its regulatory role in plant growth. Based on the recent progress achieved from the studies on flowering time responses to nitrogen (N) in Arabidopsis, we now postulate a preliminary network of nitrate signaling on flowering time in this review. However, many key questions remain open.

A link between nitrate and other signals to regulate plant growth has been actively proposed. A good example is that nitrate signaling influences plant growth by interacting with hormone networks, such as auxin [108] or gibberellin (GA) [91,109,110]. Another example here is that N signaling as an input to the central clock interferes with circadian rhythms [73]. Consequently, it alters circadian output pathways, such as changes in flowering time [65]. However, our understanding of the mechanisms by which N availability affects either GA or the circadian clock is fragmentary and largely based on transcriptional data. The question is therefore: what are the underlying molecular mechanisms involved in the interaction between the nitrate signal and other internal or external signals? Moreover, other N-form molecules may be involved in a variety of biological processes, such as ammonium [111], glutamine [112], polyamines [113,114]. The interaction between nitrate signaling and these molecules on plant growth remains an open question.

Plants can directly sense the nitrate status and mediate flowering in the shoot apical meristem (SAM) [97]. This raises the question of how spatial and temporal nitrate signaling is activated to mediate flowering time. In recent years, cell-specific transcriptomes, cell-specific epigenomics, and the development of precise fluorescent reporters have facilitated the uncovering of spatial regulatory networks. The use of these refined approaches will help us to unravel the events involved in the spatio-temporal specificities of nitrate signaling in regulating flowering time.

A full understanding of how nitrate signaling output pathways are linked to reproductive processes will allow us to take advantage of these crucial components. Together with advances in the areas of genome editing and synthetic biology, it will greatly help us propose new biotechnological strategies to improve nitrogen-use efficiency in response to N status, ultimately making a contribution to sustainable agriculture.

## Figures and Tables

**Figure 1 ijms-25-05310-f001:**
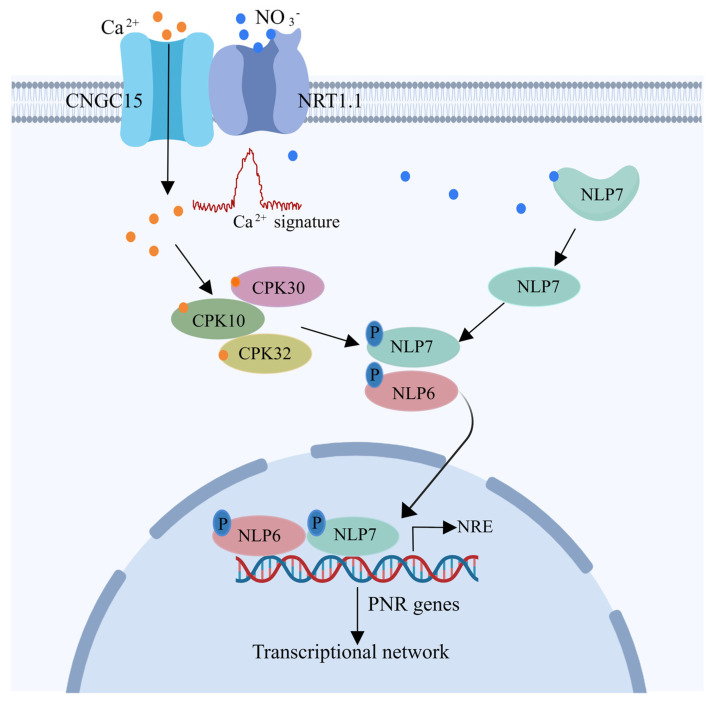
The nitrate signaling pathway in *Arabidopsis thaliana*. At elevated nitrate levels, it leads to the activation of calcium−channel activity of the NRT1.1−CNGC15 complex, resulting in a rapid nitrate−induced influx of Ca^2+^ into the cytoplasm. As cytosolic Ca^2+^ levels increase, the calcium−dependent kinases CPK10, CPK30, and CPK32 promote the phosphorylation of NLP7/NLP6. This triggers the nuclear translocation of NLPs. Meanwhile, NLP7 senses nitrate directly in the cytoplasm. Once nitrate binds to NLP7, it causes NLP7 to undergo a conformational change and to be transcriptionally derepressed. This occurs simultaneously and synergistically with the phosphorylation process. In the nucleus, NLP6 and NLP7 can bind directly to nitrate−responsive cis−elements (NRE) in the promoter regions of PNR genes and activate nitrate−responsive transcription. Abbreviations: NO_3_^−^, nitrate; Ca^2+^, calcium; P, phosphorylation; NRT1.1, NITRATE TRANSPORTER 1.1; CNGC15, CYCLIC NUCLEOTIDE-GATED CHANNEL 15; CPK10, CALCIUM-DEPENDENT PROTEIN KINASE 1; CPK30, CALCIUM-DEPENDENT PROTEIN KINASE 30; CPK32, CALCIUM-DEPENDENT PROTEIN KINASE 32; NLP7, NIN LIKE PROTEIN 7; NLP6, NIN LIKE PROTEIN 6; NRE, nitrate-responsive cis-elements; PNR, primary nitrate response.

**Figure 2 ijms-25-05310-f002:**
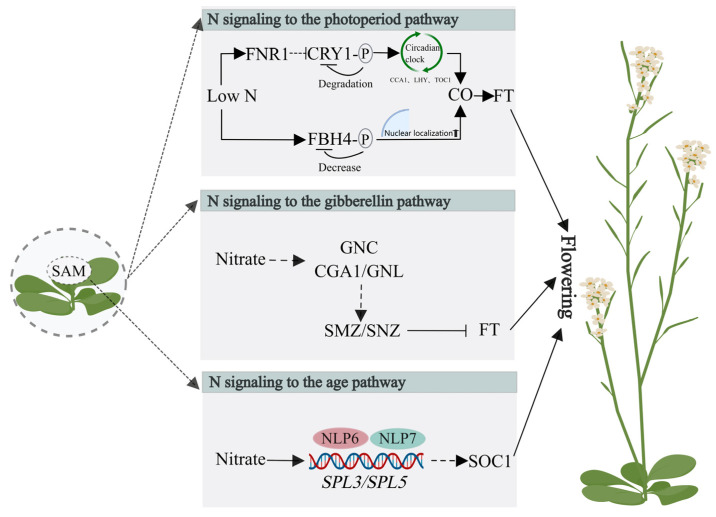
Nitrate signaling regulates flowering time in *Arabidopsis thaliana*. N signaling as an input to the central circadian clock, by modulating the cellular CRY1 protein abundance, affects the expression of key components of the central oscillator. Subsequently, as a circadian output, the expression of some flowering genes in the photoperiod pathway, such as *CO*, is altered, and this affects flowering time. Additionally, the phosphorylation state of the FBH4 protein can be modulated by N conditions. Under low N conditions, FBH4 phosphorylation levels decrease, promoting FBH4 nuclear localization and transcriptional activation of the direct target *CO* and downstream florigen *FT* to accelerate flowering. In another pathway, nitrate availability controls bolting and flowering time by modulating *SMZ* and *SNZ* gene expression via the GA pathway. In the third pathway, nitrate signaling directly integrates into the age pathway in a tissue-specific manner. NLP7 and NLP6 directly regulate *SPL3* and *SPL5* at the transcriptional level in the SAM. SOC1 is considered the main flowering integrator involved in this pathway. Abbreviations: P, phosphorylation; FNR1, FERREDOXIN-NADP (+) OXIDOREDUCTASE 1; CRY1, cryptochrome blue-light receptor 1; CCA1, CIRCADIAN CLOCK ASSOCIATED 1; LHY, LATE ELONGATED HYPOCOTYL 1; TOC1, TIMING OF CAB EXPRESSION 1; CO, CONSTANS; FT, FLOWERING LOCUS T; FBH4, FLOWERING BHLH 4; GNC, GATA TRANSCRIPTION FACTOR 21; GNL/CGA1, GATA TRANSCRIPTION FACTOR 22; SMZ, SCHLAFMUTZE; SNZ, SCHNARCHZAPFEN; NLP6, NIN LIKE PROTEIN 6; NLP7, NIN LIKE PROTEIN 7; SPL3, SQUAMOSA PROMOTER BINDING PROTEIN-LIKE 3; SLP5, SQUAMOSA PROMOTER BINDING PROTEIN-LIKE 5.

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
