# Peer review of "Nitrate Signaling and Its Role in Regulating Flowering Time in Arabidopsis thaliana"

_ijms, 2024, doi:10.3390/ijms25105310_

Round 1
Reviewer 1 Report
Comments and Suggestions for Authors
The current review is beneficial for readers whose work falls under the same subject matter because it contains summaries of recently published articles. However, I have a suggestion: it is insufficient to have just two figures, even though they include the previously published data.
Additionally, the authors should provide a table with the accomplishments and summaries of a few recent works published in the last five to eight years and one diagram showing their observations based on the published studies on nitrate signaling and their role. New findings must be addressed in the figure instead of the previously reported ones.
Apart from this, the review contains good knowledge for the respondents.
Comments on the Quality of English LanguageThe English language can be improved for some words.
Author Response
We would like to express our gratitude to you for taking the time to review this manuscript and for your valuable suggestion. Given the extensive published reviews on nitrate signaling, this review will focus on the recent advances in our understanding of the perception and transduction of nitrate signals, with a particular emphasis on the nitrate-coupled calcium signal and the novel nitrate sensor NLP7. We summarized these finding in Figure 1. As suggested, we have added a table (table 1) of the detailed description of these components in the nitrate signaling cascade and their roles in the revised manuscript. Another related topic of this review is the role of nitrate signaling in regulating flowering time in Arabidopsis. Although the effect of nitrogen nutrition on flowering time has been proposed for decades, the studies on the molecular mechanisms of nitrate signaling on flowering time are less than ten years. We summarized all the published research and postulated a preliminary network as shown in Figure 2.
Reviewer 2 Report
Comments and Suggestions for Authors
Please see attached

Minor proof reading
Author Response
Thank you very much for taking the time to review this manuscript. We really appreciate your comments. Please find the detailed responses below, and the revisions are highlighted in purple in the resubmitted files.
Line 24: We have added the information on the forms and sources of nitrogen in soil system in the revised manuscript.
Line 64-66: We absolutely agree that protein names can be in lower case. For these flowering-related proteins in Arabidopsis, all letters in upper case are normally used.
Line 170-176: We have corrected. And we check the whole manuscript to make sure all the gene names are in italics. Thanks!
Line 363-369: We have added a new open question regarding the interaction between nitrate signaling and other N-containing molecules as suggested.
Reviewer 3 Report
Comments and Suggestions for Authors
Comments on ljms-2951441
This review concerns nitrate signaling transduction, especially nitrate interaction with components regulating flowering time in Arabidopsis. It describes recent and important research in this topic and also open questions needed to be solved . However the review contains many abbreviations that are not explained. It is , therefore, difficult to read and understand for researchers, which don’t work within the same subject.
To improve the review, please explain all abbreviations first time they are mentioned in each Section.
Add explanations to all abbreviations in Figures 1 and 2.
Avoid abbreviations in subtitles.
The Perspective part should be shorter.
There are also some minor concerns.
Introduction
-Instead of: “Nitrogen (N) is the most essential macronutrient in plants”, it is better mention: Nitrogen is one of the essential macronutrients in plants.
-Lines31-32. NITRATE TRANSPORTER 1.1 (NRT1.1, also known as NPF6.3, CHL1 and NRG1)
NPF6.3, CHL1 and NRG1 have to be explained, or removed.
-Line 105. Phenotypes? If you mean the nrt1.1 mutant, you should say phenotype.
Section 2
-“Biomolecule Interaction Analyzers (MST and SPR)” Please, give a reference! And also a reference to mCitrine-NLP7!
Section 3
-Line 191.The icpk mutant? Which kinase didn’t cause phosphorylation.
Figure 1.
This figure shows that nitrate in the cytosol only affects NLP7, but not NLP6, and that both are phosphorylated by the kinases. Is it a difference between NLP6 and NLP7? How is NLP6 involved? This is not shown in the figure.
Legend of Figure 1.
Explain: CNGC15, NLP6, NLP7 and NRT1.1 and PNR.
Figure 2.
SPL3, SPL5, SMZ, SNZ and FBH4 are included in this figure, and need an explanation!
Legend of Figure 2.
Explain: CGA1/GNL, CO, CRY1, FBH4, FNR1, FT, GATA, GNC, NLP6, NLP7, SMZ, SNZ, SOC1, SPL3, SPL5.

Only minor mistakes
Author Response
Thank you very much for taking the time to review this manuscript. We are very grateful for the helpful comments. The revised manuscript is greatly improved after we made changes based on your suggestions. Please see our point-by-point responses to your comments in attachment.

Round 2
Reviewer 3 Report
Comments and Suggestions for Authors
The Authors have done the necessary changes to improve the manuscript.